# Valorization of Fig (*Ficus carica* L.) Waste Leaves: HPLC-QTOF-MS/MS-DPPH System for Online Screening and Identification of Antioxidant Compounds

**DOI:** 10.3390/plants10112532

**Published:** 2021-11-21

**Authors:** Chunying Li, Meiting Yu, Shen Li, Xue Yang, Bin Qiao, Sen Shi, Chunjian Zhao, Yujie Fu

**Affiliations:** 1Key Laboratory of Forest Plant Ecology, Ministry of Education, Northeast Forestry University, Harbin 150040, China; nefujane@aliyun.com (C.L.); klp19ymt@nefu.edu.cn (M.Y.); klp18ls@nefu.edu.cn (S.L.); klp20yx@nefu.edu.cn (X.Y.); klp20qb@nefu.edu.cn (B.Q.); klp20ss@nefu.edu.cn (S.S.); 2College of Chemistry, Chemical Engineering and Resource Utilization, Northeast Forestry University, Harbin 150040, China; 3Engineering Research Center of Forest Bio-preparation, Ministry of Education, Northeast Forestry University, Harbin 150040, China; 4Collaborative Innovation Center for Development and Utilization of Forest Resources, Harbin 150040, China; 5Heilongjiang Provincial Key Laboratory of Ecological Utilization of Forestry-Based Active Substances, Northeast Forestry University, Harbin 150040, China

**Keywords:** *Ficus carica* L. waste leaves, antioxidant, HPLC-QTOF-MS/MS-DPPH, flavonoids, polyphenols

## Abstract

Fig (*Ficus carica* L.) leaves are produced each year and often disposed, resulting in a waste of resources. Fig waste leaves are rich in flavonoids, which have strong antioxidant activity; however, the variety and chemical structure of antioxidants in fig leaves have not been reported in detail. To take full advantage of fig waste leaves, antioxidant capacity of different extracts (petroleum ether, ethyl acetate, and water) was evaluated by 1, 1-diphenyl-2-picrylhydrazyl (DPPH), 2,2′-azino-bis(3-ethylbenzothiazoline-6-sulfonic) acid (ABTS), and ferric-ion-reducing antioxidant power (FRAP) methods. The results showed that flavonoids in ethyl acetate extraction had the highest content (83.92 ± 0.01 mg/g), maximum DPPH scavenging activity (IC_50_ 0.54 mg/mL), highest ABTS scavenging rate (80.28%), and FRAP (3.46 mmol/g). Furthermore, an HPLC-QTOF-MS/MS-DPPH method was developed to identify 11 flavonoids in fig waste leaves. This rapid and efficient method can not only be used for screening the antioxidant components in fig waste leaves, but also can be combined with mass spectrometry to identify the compounds with antioxidant capacity. There are three flavonoids with significant antioxidant capacity, which are 3-O-(rhamnopyranosyl-glucopyranosyl)-7-O-(glucopyranosyl)-quercetin, isoschaftoside, and rutin. The results confirmed that fig waste leaves contain a variety of antioxidant components, which contributed to increase the value of fig waste leaves as antioxidants.

## 1. Introduction

Fig (*Ficus carica* L.), as one of the earliest cultivated fruit trees, belongs to the mulberry family (Moraceae). Fig are native to the Mediterranean coast, from Turkey to Afghanistan [1], mainly grow in some tropical and temperate regions, and belong to the subtropical larch family [2]. It was introduced into China from Persia in the Tang Dynasty. Because of its low requirements on soil conditions and strong roots, it was widely cultivated in the north and south, mainly distributed in Xinjiang, Fujian, Shandong, and other places. Fig has not only a wide range of medicinal and nutritional values, but also nourishes the stomach and clears the intestines, reducing swelling and detoxification. It is often used in the treatment of anorexia, abdominal distension and abdominal pain, hemorrhoids and constipation, dyspepsia, insufficient milk, sore throat, cough, and sputum [3,4,5].

To make fig trees grow more vigorously, they are often pruned. Therefore, a large number of abandoned branches and leaves are produced every year, causing environmental pollution and waste of resources. However, fig leaves have been found to contain flavonoids, sugars, pectin, tannins, vitamin C (Vc), and trace elements [6,7]. Clinical studies have shown that fig leaves extract have anti-tumor, hypolipidemic, antioxidant, antibacterial, hypoglycemic, and other functions [8,9,10]. Because fig leaves contain a large number of flavonoids, they have a variety of pharmacological activities, which can prevent cardiovascular diseases, treat osteoporosis, treat of diarrhea, scavenge oxidative free radicals, lower blood lipids, treat of sore throat, and regulate the immune system [11,12,13,14,15,16,17]. It is reported that high amounts of flavonoids have been found in fig leaves [18], including rutin, quercetin, anthocyanin, and so on [19]. Anthocyanins enable promotion of the apoptosis of cancer cells, slow down the proliferation of cancer cells, and prevent the metastasis of cancer cells [20]. At the same time, they can effectively relieve visual fatigue and slow down the development of diabetic cataracts [21]. Quercetin and rutin have protective effects on ischemic brain injury [22]. Flavonoids not only regulate the cardiovascular system by reducing capillary permeability, lipids, and blood pressure, but also play an important role in anti-inflammatory and immune activity by affecting cell mitosis, cell–cell interaction, and cell secretion [23,24]. Therefore, making full use of fig waste leaves is an effective way to realize resource utilization.

DPPH is a stable nitrogen-centered free radical with a maximum absorption peak at 517 nm. When an antioxidant is added to DPPH solution, it forms a light-colored substance due to its single electron pairing [25]. In the evaluation of antioxidant activity of herbal extracts with complex components, the main drawback of this method is that it is impossible to know exactly which compounds in the complex mixture have antioxidant activity. Only after the separation and purification of a compound it can be known whether it has antioxidant activity. Recently, HPLC-DPPH-DAD online screening of antioxidant systems is a fast and efficient method for exploring antioxidant compounds from natural resources [26,27] and can be used for screening and identification of antioxidants in fruit wine, tea, plants, and so on [28,29,30].

HPLC-DPPH was used to screen out the antioxidant active components in different extracts of fig waste leaves. In order to further study the antioxidant chemical components, HPLC-QTOF-MS/MS-DPPH was applied to identify and analyze the antioxidant active components for the first time. By this method, 11 flavonoids were found in the fig waste leaves. Compared with the traditional DPPH assay for screening natural antioxidant active components, this method was validated to be simple and reliable [31], which can screen and identify the antioxidant active components from plants rapidly and efficiently. The purpose of this study was to explore the chemical structure and antioxidant capacity of antioxidative compounds in fig waste leaves for making full use of the fig resources.

## 2. Results and Discussion

### 2.1. Determination of Total Polyphenol Content

The total polyphenol content of different solvent extracts of fig leaves was shown in Figure 1A. It was obvious that the ethyl acetate extract showed the highest content of polyphenol compounds, which was (1.72 ± 0.01) mg/g. In contrast, the content of polyphenol compounds measured in water and petroleum ether extracts was significantly lower, which might be related to the low solubility of polyphenol in petroleum ether and water. Most polyphenols have a certain polarity. According to the principle of similar phase dissolution, solvents with similar polarity can be selected to extract them from plant materials simply and efficiently. The moderate polarity of ethyl acetate is more similar to that of polyphenols than that of water and petroleum ether.

### 2.2. Determination of Total Flavonoids Content

The total flavonoids content of different solvent extracts of fig leaves is shown in Figure 1B. The content of total flavonoids in the extracts was mainly related to different extraction solvents. The content of total flavonoids in the extracts of different solvents was determined. Total flavonoids content in the ethyl acetate part was the highest, which was (83.92 ± 0.01) mg/g, and that in the petroleum ether part was the lowest, which was (18.71 ± 0.11) mg/g.

### 2.3. Antioxidant Capacity

DPPH, ABTS, and FRAP were used to evaluate the antioxidant capacity of extracts from fig waste leaves. DPPH free radicals could be scavenged by fig leaves extracts obtained using three different extraction solvents, and the scavenging rate of DPPH free radicals gradually increased with the increase of extract concentration. The scavenging ability of ethyl acetate extract varied greatly with concentration. As is known, the lower the IC_50_ value, the higher the antioxidant activity of the antioxidants. The IC_50_ value of ethyl acetate extract was 0.54 mg/mL, which did not exceed the scavenging ability of Vc on DPPH free radical; nevertheless, the difference was not significant (*p* > 0.05). It can be seen from Figure 2A that the scavenging ability of three extracts on DPPH free radicals decreased in the following order: ethyl acetate phase > water phase > petroleum ether phase. This may be related to the fact that the ethyl acetate extract contained more flavonoids and polyphenol compounds.

It can be seen from Figure 2B that each fig leaf extract had the ability of scavenging ABTS free radicals, which was similar to that of DPPH free radicals. The changing trend of sample scavenging ability was the same as that of sample concentration. The ethyl acetate extract had the strongest scavenging rate at different concentration of three extracts and the scavenging rate was 80.28% at the concentration of 2.5 mg/mL. The ABTS free radical scavenging activity decreased in the following order: ethyl acetate phase > water phase > petroleum ether phase—which also proved that the flavonoids of fig leaves had strong antioxidant activity. Flavonoids and polyphenols contain more phenolic hydroxyl groups, which generally show antioxidant activity by reducing hydroxyl groups. They can stabilize free radicals and play an antioxidant role by providing hydrogen ions.

When the total antioxidant capacity was measured by FRAP method, the FRAP value was represented by the concentration of FeSO_4_ solution. The higher the concentration of extracts, the stronger the antioxidant activity of the substance. The total antioxidant capacity was positively correlated with the concentration of extracts. Among the three extracts, ethyl acetate phase showed the highest FRAP value (3.46 mmol/g) and the strongest reducing ability to ferric ion, which was significantly higher than that of water and petroleum ether phase. It indicated that ethyl acetate dissolved more antioxidants.

### 2.4. Flavonoid Characterization of Fig Leaves by HPLC-DAD-ESI-MS

Flavonoids from fig waste leaves were characterized by HPLC-DAD-ESI-MS. As is shown in Figure 3, 11 negative peaks of ethyl acetate extract in fig leaves were observed at 517 nm. By scanning in negative ion mode, 11 compounds were analyzed by mass spectrometry. The larger the area of the negative peak, the stronger the antioxidant activity of the compound. The chemical structures of 11 compounds are elucidated in Figure 4. Among them, 1, 6, and 7 are the main substances with high antioxidant activity. It is speculated that they are 3-O-(rhamnopyranosyl-glucopyranosyl)-7-O-(glucopyranosyl)-quercetin, isoschaftoside, and rutin, respectively.

Under the condition of negative ions, the total ion flow diagram of flavonoids in fig leaves is shown in Figure 5. Through the analysis of the information in the first-level mass spectrometry, the molecular weight of each compound can be preliminarily inferred, as shown in Table 1. By comparing the fragmentation information of the target compounds in the secondary mass spectrometry, searching the computer standard mass spectrometry database, and combining these with the references, the experimental results were comprehensively analyzed [32].

### 2.5. Radical Scavenging Capacity of Fig Flavonoids by Online HPLC-DPPH

The isolated compound reacted with DPPH in post-column, and the antioxidants in fig leaves were screened. All 11 components of the fig leaves extract have negative peaks that can be observed in DPPH free radical detection spectrum at wavelength 517 nm in a short time (in 30 min) (Figure 3). Three main peaks (1, 6, and 7) were clearly observed as the main contributors of antioxidants. The results showed that there were abundant antioxidant substances scavenging DPPH free radicals in fig leaves. When extracted with ethyl acetate, the negative peak had a better peak shape, a stable baseline, and a large signal noise ratio (SNR).

### 2.6. Identification of Flavonoid Compounds

The active components in ethyl acetate from fig leaves were identified according to other characteristics of fragments in the mass spectrometry. All 11 flavonoids in fig leaves were identified and their chemical structures were determined. Table 1 presented the retention time, MS fragmentation, molecular formula, relative content, relative antioxidative power, and literature source of compounds, detected by HPLC-QTOF-MS/MS analysis. Three of the most prominent antioxidative compounds present in fig leaves have medicinal value and can be developed into drugs with commercial value. Quercetin and its derivatives, 3-O-(rhamnopyranosyl-glucopyra-nosyl)-7-O-(glucopyranosyl)-quercetin, showed anti-inflammatory and neuroprotective effects [44]. Isoschaftoside has a good acetylcholinesterase (AChE)-inhibition activity and is expected to become a treatment for Alzheimer’s disease [45]. As a natural antioxidant, rutin has a wide range of pharmacological activities, such as anti-tumor, anti-inflammatory, antiviral, and so on [46]. The first-level and secondary mass spectrometry of 11 flavonoids are shown in Figure 6.

## 3. Materials and Method

### 3.1. Chemicals and Materials

Discarded fig leaves were picked from Chengshan Town, Rongcheng City, Shandong Province, China, from Sep. 6 to Sep. 10. The fig variety was Bulanruike. The fig green leaves were dried at a shady place out of direct sunlight until the weight remained constant. The dried leaves were smashed and sieved through a 60-mesh sieve for further testing. All chemicals were of analytical grade, unless stated otherwise, and were purchased from Sigma Aldrich (St. Louis, MO, USA). Standard solutions were stored at 4 °C.

### 3.2. Instrumentation

An Agilent 6530 Accurate-Mass QTOF-MS system was connected with HPLC system via an Agilent Jet Stream electrospray (ESI) interface and an Agilent 1260 HPLC with diode array detector (DAD) (Agilent Technologies, Santa Clara, CA, USA). Kq-100e Ultrasonic cleaning instrument (KQ-250DB, Kunshan Ultrasonic Instruments Co., Ltd., Kunshan, China) was used.

### 3.3. Sample Preparation

The fig leaves waste were dried 24 h at the temperature of 60 °C and then crushed through a 60-mesh screen. A measure of 10 g of dry powder was weighed and put into a triangular flask; 70% ethanol was added at the ratio of 1:10 (*w*/*v*), 200 W, 40 °C; ultrasonic for for 30 min, repeated twice; centrifugation at 6000 rpm for 10 min (TG16-W, Hunan Xiang Yi Laboratory Instrument Development Co., Ltd., Changsha, China); filtrate was combined; concentration was performed to obtain the ethanol extract of fig leaves. An appropriate amount of distilled water was added to the ethanol extract to form a homogeneous suspension, which was extracted by adding petroleum ether and ethyl acetate, in turn, to be extracted by rotary evaporation at 60 °C to obtain the extracts.

### 3.4. Total polyphenol Content

The total polyphenol contents in different solvent extracts were based on the Folin–Ciocalteu method. The optimal reaction conditions of the system were as follows: test sample 0.1 mL, Folin–Ciocalteu reagent 0.1 mL, and 60 g/L Na_2_CO_3_ solution 0.8 mL, with minor modifications [47]. Mixture was incubated at room temperature and protected from light for 10 min. Absorbance of total polyphenol at 765 nm was determined by UV-Visible spectrometer (UV-2600, Shimadzu Instruments Co., Ltd., Suzhou, China).

### 3.5. Total Flavonoid Content

The content of total flavonoids in fig waste leaves was determined by NaNO_2_-Al(NO_3_)_3_ method [48]. Rutin was used as a standard. Content of total flavonoid was measured at 506 nm of wavelength.

### 3.6. DPPH Radical Scavenging Capacity

A proper amount of DPPH sample was weighed and anhydrous ethanol was added for ultrasonic dissolution to obtain DPPH solution with the concentration of 0.06 mg/mL. A measure of 0.15 mL of DPPH solution from the stock was mixed with 0.17 mL of diluted sample. The reaction was kept from light for 30 min at room temperature. The detection wavelength was 517 nm. Each group of experiments is parallel for three times, and the calculation formula was as follows:(1)$$W=\frac{1-\left(A_{m}-A_{n}\right)}{A_{k}}\times 100$$
where *W* (%) represents DPPH radical scavenging rate. *A*_k_, *A*_m_, and *A*_n_ represent absorbance of DPPH solution, absorbance of extracts of different concentrations reacted with DPPH solution, and absorbance of different concentrations of extracts reacted with anhydrous ethanol, respectively. 

### 3.7. 2,2′-Azino-bis(3-ethylbenzothiazoline-6-sulfonic) Acid (ABTS) Radical Scavenging Capacity

The mixture of 100 µL ABTS solution and oxidant solution was added at the volume ratio of 1:1 and was kept away from light at room temperature for 16 h. The ABTS working solution was obtained by diluting the ABTS solution with anhydrous ethanol. 20 µL of different concentrations of extract solution, different concentration of Vc solution, and 180 µL ABTS working solution were mixed evenly in a 96-well plate. The absorbance was measured at 734 nm, and the experiments were repeated three times in each group. The scavenging capacity of ABTS free radicals was calculated by the following Formula (2):(2)$$W=\frac{1-\left(A_{m}-A_{n}\right)}{A_{k}}\times 100$$
where *W* (%) refers to ABTS radical scavenging rate. *A*_k_, *A*_m_, and *A*_n_ refer to absorbance of ABTS working solution, absorbance of ABTS after reacting with sample, and absorbance of sample solution mixed with anhydrous ethanol, respectively.

### 3.8. Ferric-Ion-Reducing Antioxidant Power (FRAP)

FRAP was determined according to the method of Jin et al. [48]. We took 150 µL of appropriately diluted extract of fig waste leaves, added 3 mL of 2,3,5-triphenyl-2h-tetrazolium,chloride (TPTZ) working solution—which consists of acetic acid buffer (pH 3.6), 10 mmol/L TPTZ solution and 20 mmol/L FeCl_3_ solution mixed in proportion (10:1:1)—and reacted this at 37 °C for 30 min. The detection wavelength was 593 nm.

### 3.9. HPLC-DAD-ESI-MS Analysis

HPLC-ion trap mass spectrometry with DAD was used to determine the components of fig waste leaves extract. The mobile phase was consisted of 0.1% formic acid solution (A) and acetonitrile (B). The gradient elution conditions were as follows: 0–6 min, 15% B; 6–30 min, 15–45% B; 30–40 min, 45–60% B. The column was kept at 30 °C and detected at 330 nm. An Agilent 6530 Accurate-Mass QTOF-MS system was connected with HPLC system via an Agilent Jet Stream electrospray (ESI) interface (Agilent Technologies, Santa Clara, CA). The electrospray ion source was in negative ion mode, the scanning range of mass spectrometer was from *m*/*z* 100 to 1500, the ion source was set at 550 °C, the ion source voltage (IS) was −4500 V, the atomization gas pressure was 55 psi, and the air curtain gas (CUR) pressure is 35 psi. The de-clustering voltage (DP) of the first-level scanning and the focusing voltage (CE) was 100 V and 10 V, respectively. The secondary mass spectrometry scan used Product Ion-IDA mode to collect mass spectrum data, and the CID energy was set at −20, −40, and −60 V, respectively. Before injection, the CDS pump was used to correct the mass axis, so that the error of the quality axis was less than 2 ppm.

### 3.10. Online HPLC-DPPH Analysis

Antioxidants in the fig waste leaves extract were screened by online HPLC-DPPH. The flow chart of online HPLC-DPPH screening system is shown in Figure 7. Antioxidant capacity was evaluated through the negative peaks produced by the reaction of antioxidants with DPPH radicals. Separation was achieved on Agilent 1260 HPLC (Agilent Technologies infinity, Santa Clara, CA, USA). All HPLC-DPPH separation steps were carried out on a C_18_ column (4.6 mm × 250 mm, 5 μm, WatersCrop, Milford, MA, USA). The flow rate of HPLC-separated analytes and 50 µg/mL DPPH solution was set at 0.7 mL/min and 0.5 mL/min, respectively. The column temperature was maintained at 35 °C, the detection wavelength was 330 nm, and inject volume was 10 μL. The online HPLC-DPPH analysis conditions were the same as mentioned above. The sample was reacted with DPPH solution in PEEK tube (10 m × 0.25 mm), determined at 521 nm, the absorption of the compound with antioxidant activity was reduced by pairing with single electron of DPPH with a negative peak.

## 4. Conclusions

In this study, online HPLC-QTOF-MS/MS-DPPH method was used for the first time to screen and identify 11 antioxidant active components in fig waste leaves, including 3-O-(rhamnopyranosyl-glucopyranosyl)-7-O-(glucopyranosyl)-quercetin (1); 2-carboxyl-1, 4-naphthohydroquinone-4-O-glucopyranoside (2); luteolin 6-C-glucopyranoside, 8-C-arabinopyranoside (3); schaftoside (4); isoorientin (5); isoschaftoside (6); rutin (7); 2″-O-rhamnosylvitexin (8); isovitexin (9); isoquercetin (10); kaempferol-3-O-rutinoside (11). The antioxidant capacities of different extracts were based on DPPH and ABTS free radical scavenging rate and FRAP reduction ability. The ethyl acetate extract had the strongest antioxidant capacity. Furthermore, through online HPLC-DPPH analysis, compounds (1), (6), and (7) were considered to have significant antioxidant activity. Therefore, online HPLC-QTOF-MS/MS-DPPH was an effective and rapid analysis method for determining the antioxidant capacity of fig waste leaves, which provided the data support for development and utilization of fig resources.

## Figures and Tables

**Figure 1 plants-10-02532-f001:**
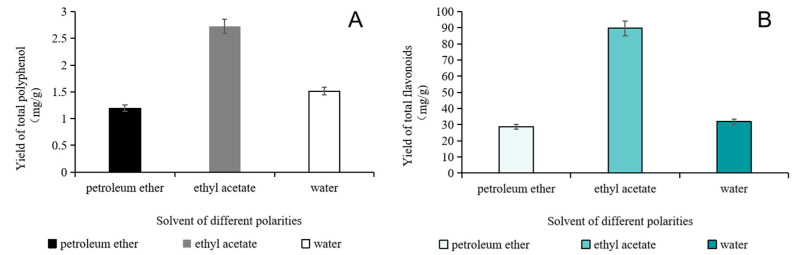
Effect of different extraction solvents on the yield of total polyphenol (**A**) and flavonoids (**B**).

**Figure 2 plants-10-02532-f002:**
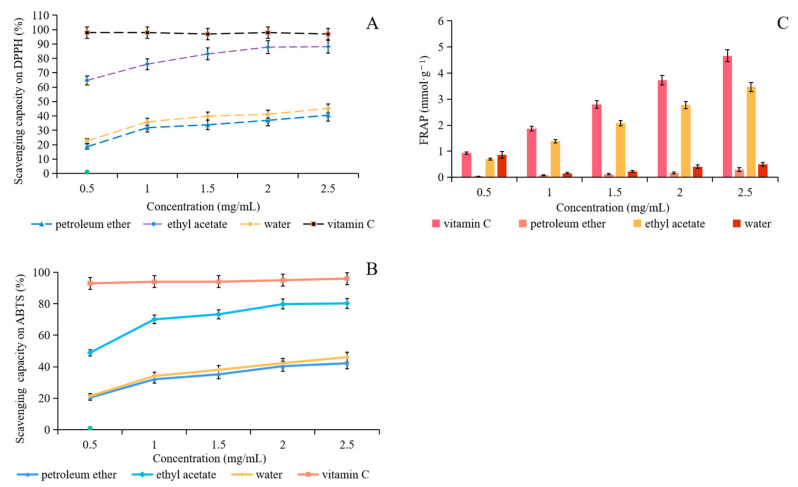
Effect of different extraction solvent on antioxidant capacity. (**A**) Scavenging capacity on DPPH radical, (**B**) scavenging capacity on ABTS radical, (**C**) ferric-ion-reducing capacity.

**Figure 3 plants-10-02532-f003:**
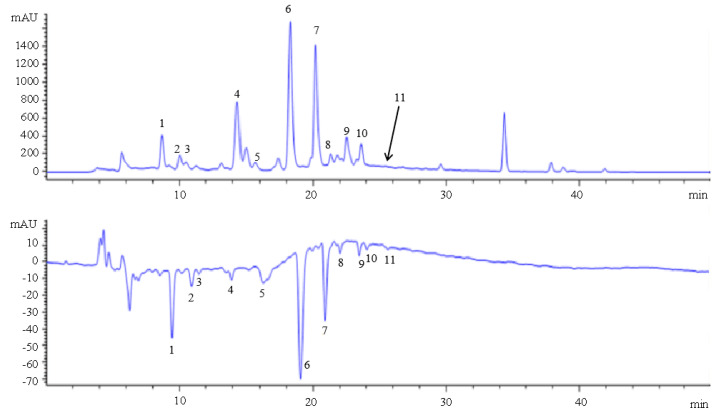
Antioxidant components of ethyl acetate extract from fig leaves by online HPLC-DPPH. 1. 3-O-(rhamnopyranosyl-glucopyranosyl)-7-O-(glucopyranosyl)-quercetin; 2. 2-carboxyl-1, 4-naphthohydroquinone-4-O-glucopyranoside; 3. luteolin 6-C-glucopyranoside, 8-C-arabinopyranoside; 4. schaftoside; 5. isoorientin; 6. isoschaftoside; 7. rutin; 8. 2″-O-rhamnosylvitexin; 9. isovitexin; 10. isoquercetin; 11. kaempferol-3-O-rutinoside.

**Figure 4 plants-10-02532-f004:**
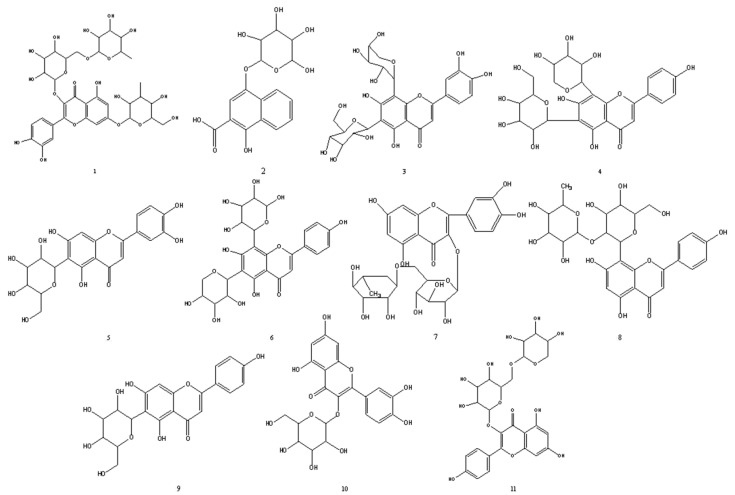
Chemical structures of 11 compounds in fig leaves. 1. 3-O-(rhamnopyranosyl-glucopyranosyl)-7-O-(glucopyranosyl)-quercetin; 2. 2-carboxyl-1, 4-naphthohydroquinone-4-O-glucopyranoside; 3. luteolin 6-C-glucopyranoside, 8-C-arabinopyranoside; 4. schaftoside; 5. isoorientin; 6. isoschaftoside; 7. rutin; 8. 2″-O-rhamnosylvitexin; 9. isovitexin; 10. isoquercetin; 11. kaempferol-3-O-rutinoside.

**Figure 5 plants-10-02532-f005:**
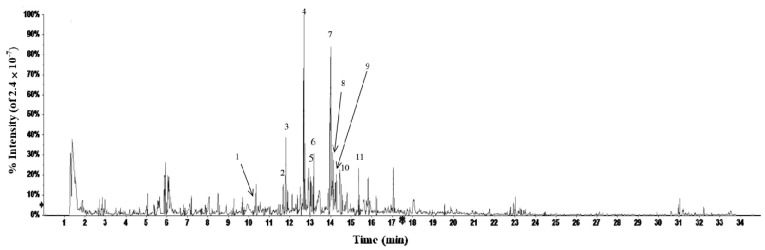
Total ion flow chromatogram of sample extract. Peaks 1–11 are identified in Table 1.

**Figure 6 plants-10-02532-f006:**
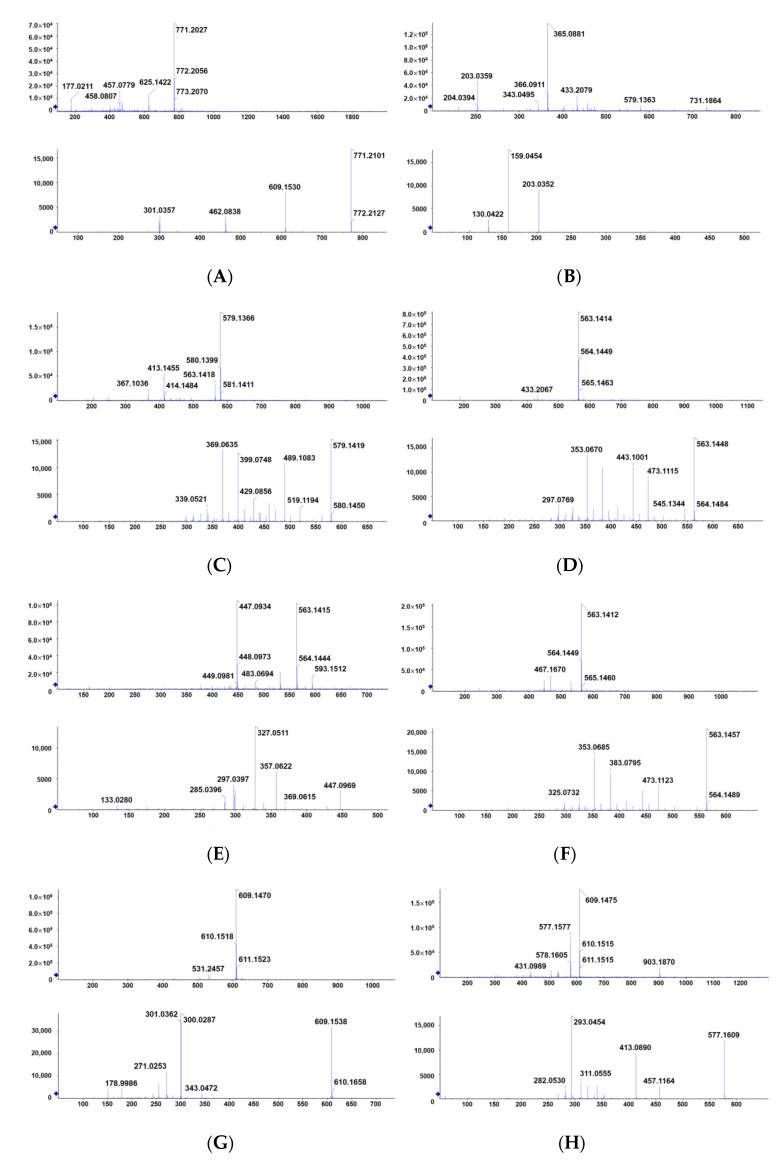
The mass spectrometry of flavonoids compounds. (**A**) 3-O-(rhamnopyranosyl-glucopyranosyl)-7-O-(glucopyranosyl)-quercetin; (**B**) 2-carboxyl-1, 4-naphthohydroquinone-4-O-glucopyranoside; (**C**) luteolin 6-C-glucopyranoside, 8-C-arabinopyranoside; (**D**) schaftoside; (**E**) isoorientin; (**F**) isoschaftoside; (**G**) rutin; (**H**) 2″-O-rhamnosylvitexin; (**I**) isovitexin; (**J**) isoquercetin; (**K**) kaempferol-3-O-rutinoside.

**Figure 7 plants-10-02532-f007:**
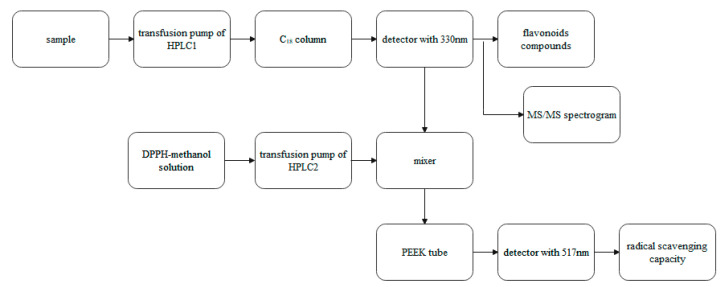
The flow chart of online HPLC-DPPH screening system.

**Table 1 plants-10-02532-t001:** Flavonoids identified in the fig leaves extracts using HPLC-QTOF-MS/MS.

PeakNo.	Retention Time (min)	First-Level Mass Spectrometry	Secondary Mass Spectrometry	Molecular Weight	Molecular Formula	Identification	Literature Resource	Relative Content ^a^ (%)	Relative Antioxidative Power ^b^
1	10.3	771	609, 462, 301	772	C_33_H_40_O_21_	3-O-(rhamnopyranosyl-glucopyranosyl)-7-O-(glucopyranosyl)-quercetin	[33]	6.4	1.7
2	11.7	365	203, 159, 130	366	C_17_H_18_O_9_	2-carboxyl-1, 4-naphthohydroquinone-4-O-glucopyranoside	[34]	3.1	1.2
3	11.8	579	519, 489, 429, 369	580	C_26_H_28_O_15_	luteolin 6-C-glucopyranoside, 8-C-arabinopyranoside	[35]	1.9	0.4
4	12.7	563	473, 443, 353	564	C_26_H_28_O_14_	schaftoside	[36]	15.6	0.1
5	13.0	447	369, 357, 327, 297, 285, 133	448	C_21_H_20_O_11_	isoorientin	[37]	1.1	7.5
6	13.2	563	443, 353, 473, 383	564	C_26_H_28_O_14_	isoschaftoside	[38]	34.4	1.0
7	14.0	609	301, 151, 257, 273	610	C_27_H_30_O_16_	rutin	[39]	27.1	0.5
8	14.2	577	457, 293	578	C_27_H_30_O_14_	2″-O-rhamnosylvitexin	[40]	1.9	0.7
9	14.3	432	341, 311, 283	432	C_21_H_20_O_10_	isovitexin	[41]	4.7	0.4
10	14.4	463	301, 151	464	C_21_H_20_O_12_	isoquercetin	[42]	3.6	0.4
11	15.4	593	285	594	C_27_H_30_O_15_	kaempferol-3-O-rutinoside	[43]	0.1	5.8

^a^ Relative content (%) represents the ratio of each positive peak area to the total positive peak area. ^b^ Relative antioxidative power represents the ratio of each negative peak area to its corresponding positive peak area.

## Data Availability

Data is contained within the article.

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
