# Peer review of "Valorization of Fig (Ficus carica L.) Waste Leaves: HPLC-QTOF-MS/MS-DPPH System for Online Screening and Identification of Antioxidant Compounds"

_plants, 2021, doi:10.3390/plants10112532_

Round 1

Reviewer 1 Report

The article is suitable for the plants journal. Some minor corrections are required to improve the article. Please find comments inside the PDF file.

Reviewer 2 Report

The authors demonstrate that HPLC-DPPH can be used to identify antioxidative compounds in fig leaves. While the presence of such compounds could enhance the value of a waste product, the authors have not defined the stage of leaf development at which the compounds were extracted or determined. All we know is that they took discarded leaves, dried them, then ground and extracted them. Were the leaves green, yellow or brown? Were they sampled in the spring, summer or fall? What variety or varieties of fig were sampled? Season, variety and especially color will determine the type and amount of secondary compounds present. This information should be provided in the article. Even better would be if the authors determined such compounds in two varieties of fig or at two different times in the season or in two defined colors of leaf.

Figure 2 define Vc

Fig 3 Please number the peaks in the top AND bottom graphs so that we can see which compounds had which degree of presence and of antioxidative power

Table 1, and in general: reduce the number of numerals after the decimal. For clarity in tables, if there is no number to the left of the decimal, up to two numbers can be to the right of the decimal. Up to two to the left of the decimal allows one number to the right of the decimal. More than two numbers to the left-- nothing to the right of the decimal (0.1 will not add anything meaningful to 1247, or to 127).

lines 184-239 and Figure 6. The words repeat themselves too much and the graphs are impossible to read. Please turn the words and the figure into a table with the name of the compound, the major and minor peaks listed, antioxidative power and the literature source. Then discuss the commercial or medicinal value of the three most prominent antioxidative compounds present in fig leaves.  

I have written major revision, but I mean moderate revision, for which there is no button to push.

Reviewer 3 Report

The manuscript titled “Valorization of Fig (Ficus carica Linn.) waste leaves: HPLC-QTOF-MS/MS-DPPH system for online screening and identification of antioxidant compounds” brings an interesting methodology to identify antioxidant compounds in agricultural by-products. The manuscript has undeniable scientific quality. However, some issues need to be fixed before publishing. Therefore, I recommend minor revisions, as outlined below:

Comment: Grammar should be revised throughout the whole manuscript.

Line 13: replace “thrown away as wastes” for “disposed.”

Line 15: not sure what the authors meant by “antioxidant components are not clear.”

Comment: many of the keywords are already contained in the title. They should be replaced to increase the article’s searchability.

Line 71: “results are not accurate.” Based on what evidence?

Line 260: dried for how long?

Item 3.3: equipment details were not given. The same goes for other items.

Item 2.1: Why are polyphenols more soluble in ethyl acetate than water and petroleum ether? This needs to be better explained.

Line 122: Bind to free radicals? Are the authors sure that this information is correct? The stabilization of free radicals happens through hydrogen donation.

Line 133: What type of chemical components?

Reviewer 4 Report

This manuscript is suitable for this journal.

The abstract, introduction are no problem. However, I suggest to have a hypothesis or study goal at the end paragraph of the introduction. The methods were designed properly to answer the research question. Those figures resolution are not enough. You may want to improve those.

In the introduction or in the discussion, you should emphasize the advantages of using the online HPLC-QTOF-MS/MS-DPPH method.
